

# Space-like dynamics in a reversible cellular automaton

**Katja Klobas⋆ and Tomaž Prosen**

Department of Physics, Faculty of Mathematics and Physics,
University of Ljubljana, Ljubljana, Slovenia

⋆ katja.klobas@fmf.uni-lj.si

## Abstract

In this paper we study the space evolution in the *Rule 54 reversible cellular automaton,* which is a paradigmatic example of a deterministic interacting lattice gas. We show that the spatial translation of time configurations of the automaton is given in terms of local deterministic maps with the support that is small but bigger than that of the time evolution. The model is thus an example of space-time dual reversible cellular automaton, i.e. its dual is also (in general different) reversible cellular automaton. We provide two equivalent interpretations of the result; the first one relies on the dynamics of quasi-particles and follows from an exhaustive check of all the relevant time configurations, while the second one relies on purely algebraic considerations based on the circuit representation of the dynamics. Additionally, we use the properties of the local space evolution maps to provide an alternative derivation of the matrix product representation of multi-time correlation functions of local observables positioned at the same spatial coordinate.

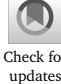

# 1 Introduction

Studying exactly solvable models has been traditionally a fruitful approach towards explaining the emergence of macroscopic phenomena from microscopics [1, 2]. In recent years, the many facets of integrability and solvability have been explored outside equilibrium physics [3]; for example, by studying long-time asymptotics of the initial value problem for a many-body interacting system — the so-called quenches [4], by developing generalized hydrodynamic description of integrable systems [5, 6], or by analysing random matrix models with intrinsic spatial locality structure — e.g. random local quantum circuits [7–10].

However, exact solutions of dynamical many-body problems for individual interacting systems are extremely scarce. A particularly interesting class of local quantum circuits that are exactly solvable in the statistical sense, yet they are not Bethe-ansatz or Yang-Baxter integrable, are *dual unitary* quantum circuits [11]. These are local interacting models in discrete space and discrete time where the roles of space and time can be exchanged while keeping dynamics unitary (a similar space-time duality has been explored in integrable field theories [12–14]). This property implies a nontrivial structure that enables exact computation of numerous physical quantities, such as local correlation functions [11, 15], entanglement spreading [16–18], operator entanglement [19, 20], and OTOCs [21]. However, dual unitarity restricts the growth of correlations to the maximal speed, which enforces strictly ballistic transport of conserved charges if present [11, 19]. Therefore these models cannot describe generic behaviour of systems with sub-ballistic transport of conserved quantities.

This motivates us to study the effect of exchanging time and space evolution in other $1+1$ dimensional models, where the strict dual unitarity condition does not hold in hope of finding a generalized space-time duality allowing for potentially richer macroscopic physical properties, such as diffusive or super-diffusive transport. This question can be rephrased in the context of classical deterministic interacting lattice systems, where the property analogous to unitarity is the symplectic feature of the dynamical evolution law. An example of dual symplectic classical lattice dynamics with continuous local degrees of freedom that exhibits super-diffusive transport in the Kardar-Parisi-Zhang universality class has been recently proposed [22, 23].

However, one can consider an even simpler class of interacting lattice systems where the local field variable takes only a discrete set of values, the so-called cellular automata. There, the feature analogous to symplecticity is the reversibility of dynamics, meaning that the local

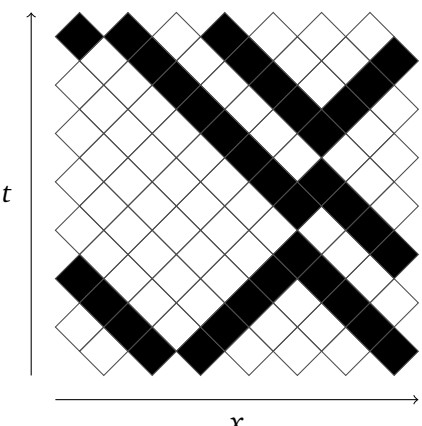

Figure 1: Example of RCA54 time evolution. The configuration at the bottom is evolved upwards according to the time evolution rules (2). Full sites (black rectangles) can be thought of as solitons that move with velocities ±1 and scattering displaces them one site backwards. If the roles of space and time are exchanged, the dynamics can be still interpreted as solitons moving with velocities ±1, but when scattering their positions are moved one site *forward* with respect to the original trajectories.

dynamical map over a discrete set of configurations is always one-to-one. A particularly interesting solvable example of such models is the Rule 54 reversible cellular automaton (RCA54) introduced by Bobenko et. al. in [24] [1] and studied extensively in the last years, both in classical [27–32] and in quantum setting [33–36]. In particular, dynamical structure factor of this model has been computed exactly [31] and shown to exhibit diffusive transport.

We argue that RCA54 is an excellent candidate for studying deterministic space evolution. Indeed, a recent study revealed that probability distributions of time configurations exhibit an efficient matrix-product description [32], suggesting that translating a given time configuration in space might be given by a composition of deterministic maps with a finite support. Another indication that a reversible space evolution formulation of RCA54 should be possible comes from the quasi-particle interpretation of the dynamics; RCA54 rules describe solitons (kinks) that move with fixed velocities and interact pairwise acquiring a delay for one site after each scattering. Exchanging the roles of space and time results in similar dynamics, the only difference is that the scattering now moves the solitons one site forward with respect to the original trajectory. See Figure 1 for a representative example.

In the paper we put this intuitive picture on formal grounds by expressing the space evolution in terms of local deterministic maps, i.e. again as a reversible cellular automaton. In Section 2 we define the model and introduce the statistical states and observables. In Section 3 we construct the local space evolution rules. We demonstrate that the space evolution is indeed local and deterministic, but it has to be defined on a reduced configuration space since not all of the configurations can be realized in the time evolution. Therefore, the corresponding spatial cellular automaton is configurationally constrained. Furthermore, the map implementing the space evolution has a larger support than the local time evolution map. In Section 4 we provide an alternative view of the problem by recasting the dynamics in terms of

---

[1]The model should not be mistaken for the cellular automaton given by rule 54 according to Wolfram's classification [25] which is not reversible. Even though both systems have the same local update rule, the way the update is implemented completely changes the dynamics. In particular, such reversible cellular automata can be understood as a caricature of 2nd order differential equations (2nd Newton's law) rather than 1st order (rate equation). This point is made more explicit by Takesue's classification [26] under which RCA54 can be interpreted as ERCA250R.

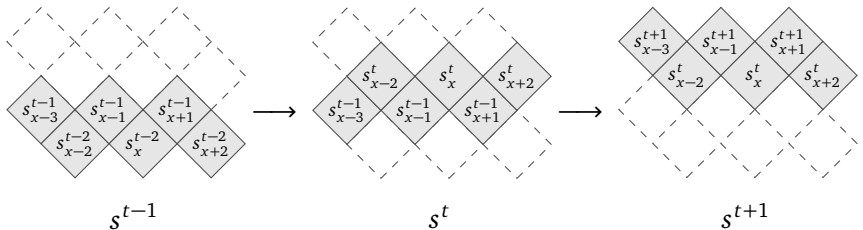

Figure 2: Schematic representation of the lattice geometry and time evolution. At every time step, half of the sites are updated as expressed in (3). The new value depends on the values of the three consecutive sites as described in (2).

reversible logical circuits with three-site gates. This allows us to express the space evolution in an equivalent but simpler way. In Section 5 we use the circuit representation to find an alternative construction of *time-states* (as introduced in [32]) that does not explicitly depend on the quasi-particle interpretation. Finally, Section 6 contains some closing remarks.

## 2 The model

### 2.1 Rule 54 dynamics

The model is defined on a one-dimensional zig-zag lattice of even length $2n$ with each site being either occupied or empty. A configuration at time $t$ is given as a string of $2n$ binary digits, $\underline{s}^t = (\ldots, s_x^t, s_{x+1}^{t-1}, s_{x+2}^t, \ldots)$, where the subscript denotes the position coordinate along the chain, superscript is the time coordinate and the time and space coordinates have the same parity, $x + t \equiv 0 \pmod 2$. Explicitly,

$$\underline{s}^{2t} = (s_1^{2t-1}, s_2^{2t}, s_3^{2t-1}, \ldots, s_{2n}^{2t}), \qquad \underline{s}^{2t+1} = (s_1^{2t+1}, s_2^{2t}, s_3^{2t+1}, \ldots, s_{2n}^{2t}), \tag{1}$$

where $s_x^t = 1$ represents an occupied site and $s_x^t = 0$ an empty one. The time evolution is defined in discrete time and it is characterized by a local three site update rule that changes the value of the middle bit (site) depending on the configuration of the triple of neighbouring sites,

$$s_2' = \chi(s_1, s_2, s_3) = s_1 + s_2 + s_3 + s_1 s_3 \pmod 2. \tag{2}$$

At every time step, the bits with the smaller time label are updated,

$$s_x^{t+1} = \chi(s_{x-1}^t, s_x^{t-1}, s_{x+1}^t), \tag{3}$$

where the *periodic boundaries* are assumed, $s_{2n+1}^t \equiv s_1^t$. Geometrically, the time evolution can be imagined to update the bottom sites of the zig-zag chain upwards while the previous top sites become the bottom sites of the propagated chain corresponding to the new time step, as schematically shown in Figure 2. Using this convention, the local time evolution rule (2) can be represented graphically as

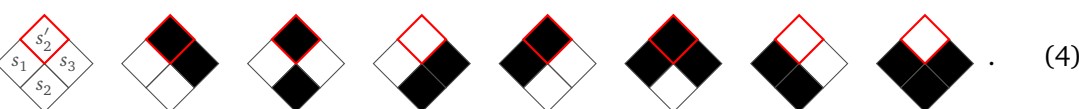 . (4)

This graphic representation of update rules immediately offers an alternative interpretation of the dynamics. Black sites represent particles that move with a constant velocity 1 either to the left or right. When two oppositely moving particles meet, they annihilate each other and

reappear in the next time step continuing with the same velocity. Or, alternatively speaking, they form a virtual bound state which decays after one unit of time. As a result, their positions are shifted backwards by one site with respect to the original trajectories. This behaviour can be also observed by considering the example of time evolution shown in Figure 1.

## 2.2 Macroscopic states

The statistical states of the system are probability distributions over the configuration space. They can be represented as vectors from $\mathbb{R}^{2^{2n}}$ with an appropriate normalization,

$$\mathbf{p} = \begin{bmatrix} p_0 & p_1 & \cdots & p_{2^{2n}-1} \end{bmatrix}^T, \qquad \sum_{s=0}^{2^{2n}-1} p_s = 1, \tag{5}$$

where the component $p_s \geq 0$ corresponds to the probability of the configuration $(s_1, s_2, \ldots, s_{2n})$ given by the binary representation of $s$; $s = \sum_{j=1}^{2n} 2^{2n-j} s_j$.[2] The time evolution of statistical states is given in terms of a local three-site permutation operator $U$ that leaves the left and right sites intact, while the middle site is changed according to the update rule (2),

$$U = \begin{bmatrix} 1 & & & & & & & \\ & & 1 & & & & & \\ & & & 1 & & & & \\ & 1 & & & & & & \\ & & & & & 1 & & \\ & & & & & & 1 & \\ & & & & 1 & & & \\ & & & & & & & 1 \end{bmatrix}, \qquad U_{(s_1',s_2',s_3'),(s_1,s_2,s_3)} = \delta_{s_1',s_1} \delta_{s_2',\chi(s_1,s_2,s_3)} \delta_{s_3',s_3}. \tag{6}$$

Due to the staggering, the full time evolution of states is given by the alternation of even and odd time evolution operators $U^{\mathrm{e/o}}$,

$$\mathbf{p}(t+1) = \begin{cases} U^{\mathrm{e}}\mathbf{p}(t), & t \equiv 0 \pmod 2, \\ U^{\mathrm{o}}\mathbf{p}(t), & t \equiv 1 \pmod 2, \end{cases} \tag{7}$$

where $U^{\mathrm{e}}$ ($U^{\mathrm{o}}$) are products of local operators $U$ acting on even (odd) triples of sites,

$$U^{\mathrm{e}} = \prod_{j=1}^{n} U_{2j}, \qquad U^{\mathrm{o}} = \prod_{j=1}^{n} U_{2j+1}, \tag{8}$$

with $U_k$ being the shorthand notation for the local operator $U$ that acts nontrivially on the sites $(k-1, k, k+1)$,

$$U_k \equiv \mathbb{1}^{\otimes k-2} \otimes U \otimes \mathbb{1}^{\otimes 2n-k-1}. \tag{9}$$

A distinguished set of statistical states are *the stationary states*, which are invariant under the time evolution. Due to the staggering, we require these states to map into themselves after *even* time steps and therefore each stationary state is associated with two vectors, $\mathbf{p}$ and $\mathbf{p}'$, corresponding to even and odd time steps respectively,

$$\mathbf{p}' = U^{\mathrm{e}}\mathbf{p}, \qquad \mathbf{p} = U^{\mathrm{o}}\mathbf{p}'. \tag{10}$$

We consider a simple class of 2-parameter stationary states, introduced in [29, 30]. The state, denoted by $\mathbf{p}(\xi, \omega)$, exhibits an efficient matrix product representation and is a simple

---

[2]To simplify notation, we will interchangeably use $p_{\underline{s}}$, $p_s$ or $p_{s_1,s_2,\ldots,s_{2n}}$ to denote probability of a configuration $(s_1, s_2, \ldots, s_{2n})$, depending on convenience.

example of a (*generalized*) Gibbs state. The two parameters $\xi, \omega$ are connected to the chemical potentials corresponding to the densities of left and right moving solitons [3]. To express the state, we first define $\mathbf{W}(\xi, \omega)$ and $\mathbf{W}'(\xi, \omega)$ as vectors in the physical space,

$$\mathbf{W}(\xi, \omega) = \begin{bmatrix} W_0(\xi, \omega) \\ W_1(\xi, \omega) \end{bmatrix}, \qquad \mathbf{W}'(\xi, \omega) = \begin{bmatrix} W_0'(\xi, \omega) \\ W_1'(\xi, \omega) \end{bmatrix}, \tag{11}$$

where the components $W_s^{(\prime)}$ ($s = 0, 1$) are matrices, acting on a three-dimensional auxiliary space,

$$W_0(\xi, \omega) = \begin{bmatrix} 1 & 0 & 0 \\ \xi & 0 & 0 \\ 1 & 0 & 0 \end{bmatrix} = W_0'(\omega, \xi), \qquad W_1(\xi, \omega) = \begin{bmatrix} 0 & \xi & 0 \\ 0 & 0 & 1 \\ 0 & 0 & \omega \end{bmatrix} = W_1'(\omega, \xi). \tag{12}$$

Note that the pair of matrices $W_s'(\xi, \omega)$ is obtained from $W_s(\xi, \omega)$ by exchanging the roles of the parameters. The stationary state $\mathbf{p}(\xi, \omega)$ takes the following matrix product form,

$$\mathbf{p}(\xi, \omega) = \frac{1}{Z_{2n}(\xi, \omega)} \mathrm{tr}\big(\mathbf{W}_1(\xi, \omega)\mathbf{W}_2'(\xi, \omega)\mathbf{W}_3(\xi, \omega)\cdots\mathbf{W}_{2n}(\xi, \omega)\big), \tag{13}$$

where the subscripts of bold vectors refer to the physical sites, and $Z_{2n}(\xi, \omega)$ is the partition sum,

$$Z_{2n}(\xi, \omega) = \mathrm{tr}\Big((W_0(\xi, \omega) + W_1(\xi, \omega))(W_0'(\xi, \omega) + W_1'(\xi, \omega))\Big)^n. \tag{14}$$

To lighten the notation, when not ambiguous, we will suppress the explicit dependence on the parameters.

The matrices $W_s, W_s'$ fulfill the following cubic algebraic relation,

$$W_{s_1} W_{\chi(s_1,s_2,s_3)}' W_{s_3} S = W_{s_1} S W_{s_2} W_{s_3}', \qquad s_1, s_2, s_3 = 0, 1, \tag{15}$$

where we introduced the matrix $S$,

$$S = \begin{bmatrix} 1 & 0 & 0 \\ 0 & 0 & 1 \\ 0 & 1 & 0 \end{bmatrix}, \qquad S^2 = \mathbb{1}. \tag{16}$$

The relation (15) can be compactly summarized as

$$U\mathbf{W}_1\mathbf{W}_2'\mathbf{W}_3 S = \mathbf{W}_1 S\mathbf{W}_2\mathbf{W}_3', \tag{17}$$

where each of the 8 physical components corresponds to one of the combinations of $(s_1, s_2, s_3)$ in (15). Defining the odd-time version of the state as

$$\mathbf{p}' = \frac{1}{Z_{2n}} \mathrm{tr}\big(\mathbf{W}_1'\mathbf{W}_2\mathbf{W}_3'\cdots\mathbf{W}_{2n}\big), \tag{18}$$

we can quickly see that time invariance condition (10) follows by repeatedly applying the cubic relation (17) and taking into account that the local time evolution operator is its own inverse $U = U^{-1}$ (for details see [29]).

---

[3]More concretely, $\log \xi$ and $\log \omega$ are precisely the chemical potentials corresponding to the densities of left and right movers respectively.

## 2.3 Local observables

Observables are real valued functions over the set of configurations and form a commutative algebra,

$$\mathcal{A}, \mathcal{B} : \mathbb{Z}_2^{2n} \to \mathbb{R}, \qquad (\mathcal{A}\mathcal{B})(\underline{s}) = \mathcal{A}(\underline{s})\mathcal{B}(\underline{s}). \tag{19}$$

The space of observables can be thought of as a vector space that is dual to the space of macroscopic states (probability vectors) $\mathbf{p}$. It is possible to define time evolution of observables via the following explicit expression of expectation values,

$$\langle \mathcal{A}(t) \rangle_{\mathbf{p}} = \sum_{\underline{s}^0} \mathcal{A}(\underline{s}^t) p_{\underline{s}^0}. \tag{20}$$

In the paper, however, we will mostly deal with *one-site observables*, which only depend on the configuration at one site, $\mathcal{A}_x(\underline{s}) = a(s_x)$, where $a$ is as a real valued function from the one-site configuration space, $a : \mathbb{Z}_2 \to \mathbb{R}$. Therefore, the expectation value of $a$ at site $x$ and time $t$ takes the following form,

$$\langle a(x,t) \rangle_{\mathbf{p}} = \sum_{\underline{s}^0} p_{\underline{s}^0}\, a(s_x^t). \tag{21}$$

This expression can be also interpreted as an inner product by introducing a one-site (unnormalized) maximum entropy vector $\boldsymbol{\omega}$ and a diagonal matrix representation of the observable,

$$\boldsymbol{\omega} = \begin{bmatrix} 1 & 1 \end{bmatrix}, \qquad \mathcal{O}_x(a) = \mathbb{1}^{\otimes x-1} \otimes \begin{bmatrix} a(0) & 0 \\ 0 & a(1) \end{bmatrix} \otimes \mathbb{1}^{\otimes 2n-x}. \tag{22}$$

Using this notation, the expectation values (21) can be expressed as

$$\langle a(x,t) \rangle_{\mathbf{p}} = \boldsymbol{\omega}^{\otimes 2n} \mathcal{O}_x(a) \mathbf{p}(t). \tag{23}$$

## 2.4 Time states

We proceed to define *time configurations* as configurations of empty/full sites observed at the same position $x$ and different times $t$, e.g. configurations of vertical zig-zag shaped chains from Figure 1. Analogously to (1), time configurations $\underline{s}_x$ are bit sequences

$$(\dots, s_x^{t-2}, s_{x-1}^{t-1}, s_x^t, s_{x-1}^{t+1}, \dots), \tag{24}$$

where the space and time label have the same parity, $x + t \equiv 0 \pmod 2$. Explicitly,

$$\underline{s}_{2x} = (s_{2x-1}^1, s_{2x}^2, s_{2x-1}^3, \dots, s_{2x}^{2m}), \qquad \underline{s}_{2x+1} = (s_{2x+1}^1, s_{2x}^2, s_{2x+1}^3, \dots, s_{2x}^{2m}). \tag{25}$$

For simplicity we assume that the time label $t$ takes the values in a finite range between 1 and $2m$. In analogy with the statistical states, *time states* are probability distributions over the space of time configurations and can be represented as vectors from $\mathbb{R}^{2^{2m}}$,

$$\mathbf{q} = \begin{bmatrix} q_0 & q_1 & q_2 & \dots q_{2^{2m}-1} \end{bmatrix}^T, \qquad \sum_{s=0}^{2^{2m}-1} q_s = 1, \tag{26}$$

where each component $q_s \geq 0$ corresponds to the probability of the time configuration given by the binary representation of $s$.

However, not every string of $2m$ binary digits represents a valid time configuration. In particular, the rules (4) imply that in a time configuration full sites always come in pairs, while three consecutive full sites are forbidden. Therefore it makes sense to restrict the discussion

only to *allowed* (also referred to as *accessible*) time configurations, where no substrings $(1, 1, 1)$ or $(0, 1, 0)$ appear. Accordingly, the only nonzero components of time states should correspond to allowed configurations. This is equivalent to requiring the time states to be invariant under the action of local projectors $P_k$,

$$P_k \mathbf{q} = \mathbf{q}, \qquad P_k \equiv \mathbb{1}^{\otimes k-2} \otimes P \otimes \mathbb{1}^{\otimes 2m-k-1}, \tag{27}$$

where $P$ is the 3 site projector to the allowed subspace of states,

$$P_{(s_1', s_2', s_3'),(s_1, s_2, s_3)} = \delta_{s_1', s_1} \delta_{s_2', s_2} \delta_{s_3', s_3} (1 - \delta_{s_1, s_3} \delta_{s_2, 1}). \tag{28}$$

In analogy with stationary states one can define *equilibrium time-states*. These states correspond to probability distributions of time configurations observed under the assumption of the system being in the equilibrium state $\mathbf{p}$ as introduced by Eq. (13). Explicitly, the equilibrium time-states are uniquely determined by the expectation values of *multi-time correlation functions* of one-site observables at the same position, [4] $C_{a_1, a_2, \ldots, a_{2m}}(\mathbf{p}) = \lim_{n \to \infty} C^{(2n)}_{a_1, a_2, \ldots, a_{2m}}(\mathbf{p})$, defined as the large system size limit of the following correlation function,

$$C^{(2n)}_{a_1, a_2, a_3, \ldots, a_{2m}}(\mathbf{p}) = \langle a_1(n^*, 0) a_2(n^*+1, 1) a_3(n^*, 2) \cdots a_{2m}(n^*+1, 2m-1) \rangle_{\mathbf{p}}, \\ \text{where} \quad n^* = 2 \left\lfloor \frac{n}{2} \right\rfloor, \tag{29}$$

i.e. $n^* = n$ for even $n$ and $n^* = n-1$ for odd $n$. Note that we choose $n^* = n$ to denote an even site close to the middle of the chain of length $2n$. By definition (23), the correlation function can therefore be recast as the following inner product between the vector $\boldsymbol{\omega}^{\otimes 2n}$ and the equilibrium distribution vector $\mathbf{p}$ on which the appropriate sequential product of time evolution operators $U^{\text{e/o}}$ and matrix representations of local observables $\mathcal{O}_{n^*/n^*+1}(a_j)$ is applied,

$$C^{(2n)}_{a_1, a_2, a_3, \ldots, a_{2m}}(\mathbf{p}) = \boldsymbol{\omega}^{\otimes 2n} \mathcal{O}_{n^*+1}(a_{2m}) U^{\text{e}} \cdots U^{\text{e}} \mathcal{O}_{n^*}(a_3) U^{\text{o}} \mathcal{O}_{n^*+1}(a_2) U^{\text{e}} \mathcal{O}_{n^*}(a_1) \mathbf{p}. \tag{30}$$

The equilibrium time state $\mathbf{q}$ is determined as the probability distribution that uniquely fixes all the values of multi-time correlation functions $C_{a_1, a_2, \ldots a_{2m}}(\mathbf{p})$,

$$C_{a_1, a_2, \ldots a_{2m}} = \sum_{s_1, s_2, \ldots, s_{2m}} q_{s_1 s_2 \ldots s_{2m}} \prod_j^{2m} a_j(s_j) = \boldsymbol{\omega}^{\otimes 2m} \mathcal{O}_{2m}(a_{2m}) \cdots \mathcal{O}_2(a_2) \mathcal{O}_1(a_1) \mathbf{q}, \tag{31}$$

where the first equality follows from the definition and the second equality is just the convenient vectorial representation. As was shown in Ref. [32], these equilibrium time-states exhibit a simple matrix product representation.

# 3 Space evolution

Our goal is to express the evolution of time configurations in the space direction as schematically shown in Figure 3. In general, there is no guarantee that the space evolution can be expressed as a composition of local deterministic maps. In our case, however, we expect this to be the case due to the soliton description of the model: the dynamics in the space direction can be understood as particles moving either upwards or downwards with velocity 1. When two oppositely moving particles meet, they get displaced one site forward with respect

---

[4]Due to the geometry of the problem, the observables are technically positioned at one of the two neighbouring sites, depending on the parity of the time-step.

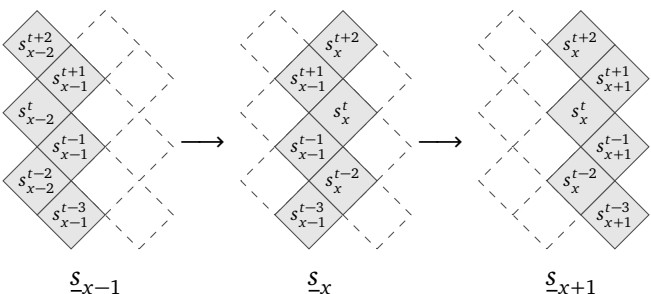

Figure 3: Illustration of the geometry of space evolution. In analogy with the time evolution shown in Figure 2, at every step the bits with the smaller space label deterministically change, while the others stay the same.

to their original trajectories, mimicking repulsive interaction. This suggests the existence of a deterministic local map,

$$s^t_{x+1} = \phi(s^{t-r}_{x/x-1}, \dots, s^{t-2}_{x-1}, s^{t-1}_x, s^t_{x-1}, s^{t+1}_x, s^{t+2}_{x-1}, \dots, s^{t+r}_{x/x-1}),\tag{32}$$

where $r \in \mathbb{N}$ characterizes the support (of size $2r+1$) of the map.

The time evolution diagrams (4) immediately imply that local space propagators cannot be expressed in terms of maps with support 3 (i.e. $r = 1$). Indeed, it is easy to see that the closest two neighbouring sites do not encode enough information to deterministically propagate the state in space. In particular, the last two pairs of diagrams have the same configurations of the left three bits and different values of the right site. Therefore, the support must be larger. Note that we additionally have to require that the local space maps shifted by an even number of sites commute, i.e. the order in which we apply (32) on a given time configuration should not matter.

It is easy to see that the support 7 (i.e. $r = 3$) suffices to express the deterministic space propagation rules. We start by observing that the first four of the diagrams (4) give the 3-site deterministic mapping also in the space direction,

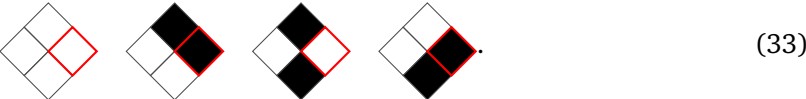 (33)

Now let us consider the subconfiguration $(0, 1, 1)$, which does not have a unique 3-site mapping and we add two neighbouring sites on the top. By avoiding the forbidden subconfigurations, there are only two possibilities of how the configuration can continue; either $(0, 1, 1, 0, 0)$ or $(0, 1, 1, 0, 1)$, which can be explicitly visualised as

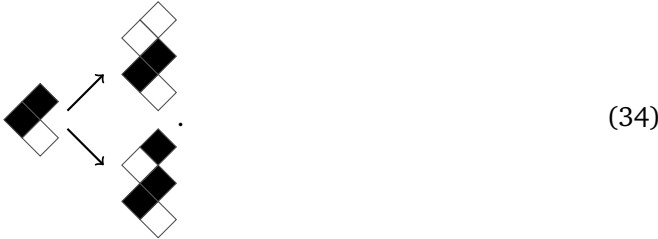 (34)

The top three sites in both configurations can be uniquely evolved by the 3-site local maps (33). After applying these deterministic rules we try to update the central bit to value 0 or 1, while requiring that the updated configuration does not violate the time-configuration restriction (27).

In both cases only one configuration obeys the restriction,

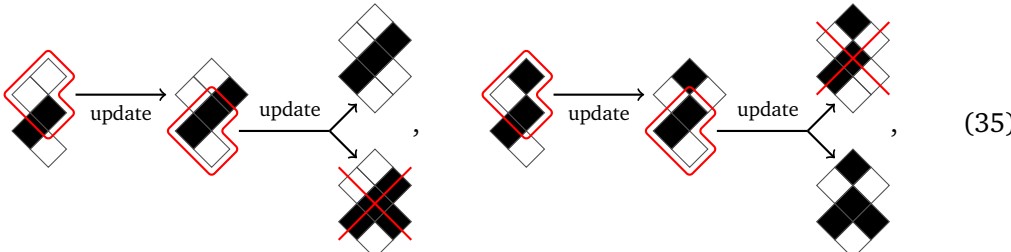

$$(35)$$

which provides a deterministic mapping corresponding to 5th and 6th diagram of the time evolution rules (4). By adding two undetermined sites to the bottom (denoted by grey squares), the two (formally 7-site) maps are graphically represented as

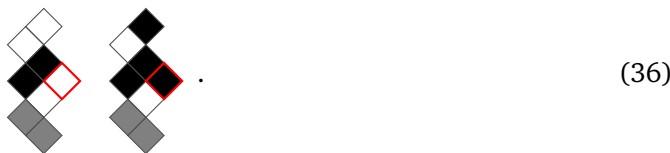

$$(36)$$

The last two rules, corresponding to 7th and 8th diagram of (4), are obtained by flipping (36) upside down,

$$(37)$$

Combining the diagrams (36) and (37) together with the simple 3-site update rules (33) completes the construction of local deterministic space evolution maps. They take the following explicit form,

$$
\phi(s_1, s_2, s_3, s_4, s_5, s_6, s_7) =
\begin{cases}
0; & s_3 = s_4 = s_5 = 0, \\
1; & s_3 = s_4 = 0, s_5 = 1, \\
0; & s_3 = 0, s_4 = s_5 = 1, s_7 = 0, \\
1; & s_3 = 0, s_4 = s_5 = 1, s_7 = 1, \\
1; & s_3 = 1, s_4 = s_5 = 0, \\
0; & s_3 = 1, s_4 = 0, s_5 = 1, \\
0; & s_1 = 0, s_3 = s_4 = 1, s_5 = 0, \\
1; & s_1 = 1, s_3 = s_4 = 1, s_5 = 0.
\end{cases}
\tag{38}
$$

Since the update rules $s_4' = \phi(s_1, s_2, s_3, s_4, s_5, s_6, s_7)$ do not depend explicitly on the values of the sites $s_2$ and $s_6$, all the local maps applied at the same step commute.

# 4 Circuit representation

## 4.1 The dual picture

Even though the local space evolution can be straightforwardly obtained by considering local subconfigurations, it requires carefulness to check all the possible cases. In this section we provide a more intuitive *circuit representation* of dynamics, which provides a simpler and more formal algebraic interpretation of local space propagation rules.

The local 3-site time evolution operator $U$ (cf. (6)) acts on three consecutive sites and changes only the value of the middle site, $U_{(s_1',s_2',s_3'),(s_1,s_2,s_3)} = \delta_{s_1,s_1'}\delta_{\chi(s_1,s_2,s_3),s_2'}\delta_{s_3,s_3'}$, which can be represented by the following graphical notation,

$$
\tag{39}
$$

Thus, $U$ is a reversible single bit gate conditioned on the values of two neighbouring bits. Using it, the full time evolution can be represented as a grid with gates centered on sites $x + t \equiv 0$ (mod 2),

$$
\tag{40}
$$

For simplicity we assume periodic boundary conditions in space and time directions. To obtain this circuit, we implicitly take into account the commutativity of gates that share at most one site: since the bit at the small circle does not change, neighbouring gates can be "merged" together into a horizontal line consisting of small and big circles. The symmetric form of (40) suggests that the dynamics can be mapped to a 2-dimensional vertex model, as described in Appendix B. Therefore, the roles of space and time can be formally exchanged by introducing the following local dual evolution operator $\hat{U}$,

$$
\hat{U}_{(s_1',s_2',s_3')(s_1,s_2,s_3)} = \delta_{s_1',s_1}\delta_{s_3',s_3}U_{(s_2,s_1,s_2'),(s_2,s_3,s_2')}.
\tag{41}
$$

Then the picture (40) can be replaced by the following

$$
\hat{U}^{\mathrm{e}}\hat{U}^{\mathrm{o}}\hat{U}^{\mathrm{e}}\hat{U}^{\mathrm{o}}\cdots
$$

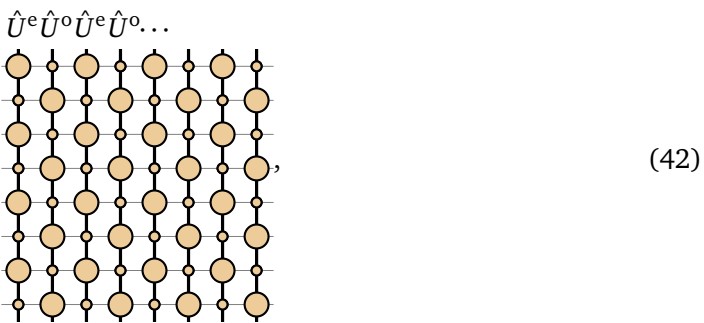

$$
\tag{42}
$$

where $\hat{U}^{\mathrm{e/o}}$ are the products of local operators acting on even/odd triplets of sites,

$$
\hat{U}^{\mathrm{e}} = \prod_t \hat{U}_{2t}, \qquad \hat{U}^{\mathrm{o}} = \prod_t \hat{U}_{2t+1}.
\tag{43}
$$

Here, $\hat{U}_t$ denotes the local operator $\hat{U}$ acting nontrivially on the triplet of sites $(t-1, t, t+1)$,

$$
\hat{U}_t \equiv \mathbb{1}^{\otimes t-2} \otimes \hat{U} \otimes \mathbb{1}^{\otimes 2m-t-1}.
\tag{44}
$$

The local operators acting on all odd (or all even) triples commute, but they are clearly not deterministic,

$$
\hat{U} =
\begin{bmatrix}
1 & & & & & & & \\
 & 0 & & 1 & & & & \\
 & & 0 & & & & & \\
 & 1 & & 1 & & & & \\
 & & & & 0 & & 1 & \\
 & & & & & 1 & & \\
 & & & & 1 & & 1 & \\
 & & & & & & & 0
\end{bmatrix} .
\tag{45}
$$

Nonetheless, as we show in the remainder of this section, projecting to the reduced space of allowed time-states (see the discussion in 2.4), space propagation can be expressed as a product of local deterministic gates with bigger support.

## 4.2 Projected dual propagators

We start by noting that the local dual operator $\hat{U}$ projects to the subspace of allowed 3-site configurations and commutes with the projector introduced in (28),

$$
\hat{U} = P\hat{U} = \hat{U}P.
\tag{46}
$$

Therefore, defining $P^{e/o}$ as the products of projectors on appropriate triples of sites,

$$
P^e = \prod_t P_{2t}, \qquad P^o = \prod_t P_{2t+1},
\tag{47}
$$

a relation similar to (46) holds for the even/odd dual propagators $\hat{U}^{e/o}$,

$$
\hat{U}^e = P^e \hat{U}^e = \hat{U}^e P^e, \qquad \hat{U}^o = P^o \hat{U}^o = \hat{U}^o P^o.
\tag{48}
$$

This allows us to represent the space evolution on the restricted space in terms of *projected dual operators* $\tilde{U}^{e/o}$,

$$
\tilde{U}^e = P^o \hat{U}^e P^o, \qquad \tilde{U}^o = P^e \hat{U}^o P^e.
\tag{49}
$$

Explicitly, by projecting to the space spanned by allowed time configurations at the beginning and at the end, the space evolution for $2m$ sites can be equivalently expressed in terms of the projected space propagators as

$$
P^e P^o \underbrace{\hat{U}^e \cdots \hat{U}^o \hat{U}^e}_{2m} P^e P^o = \underbrace{\tilde{U}^e \tilde{U}^o \cdots \tilde{U}^e}_{2m},
\tag{50}
$$

which follows directly from the definition of $\tilde{U}^{e/o}$ (49) and the commutativity of local projectors centered at different sites. This equality can be visualised graphically, by first introducing the following representation for the projector $P$,

$$
\begin{array}{l}
s_1 \;\text{———}\; s_1' \\
s_2 \;\text{——}|_P\; s_2' \\
s_3 \;\text{———}\; s_3'
\end{array} ,
\tag{51}
$$

and transforming the diagram (42) as

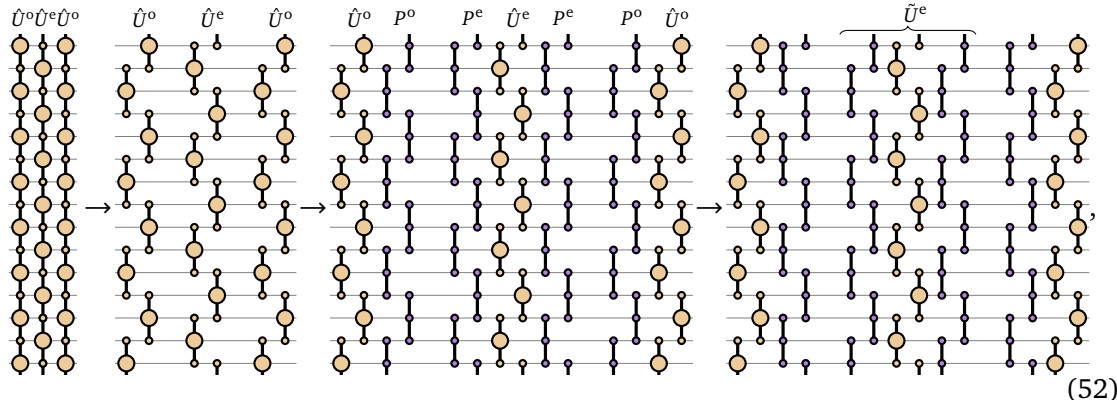

$$\tag{52}$$

where we took into account the fact that the only combinations of noncommuting gates are the ones with the big circle of one gate sitting on the same line as the small circle of another one. Explicitly, in this case the following 3 pairs *do not* commute,

$$\tag{53}$$

### 4.3 Deterministic local 7-site gates

The one-step space evolution operators $\tilde{U}^{e/o}$ can be written as products of local gates with support 7 that are deterministic on the reduced configuration subspace, i.e. it is possible to suitably define local propagators $\tilde{V}$ and $\tilde{W}$ so that $\tilde{U}^{o/e}$ take the following form,

$$
\tilde{U}^{e} = \Big(\prod_t \tilde{W}_{8t+10}\Big)\Big(\prod_t \tilde{W}_{8t+6}\Big)\Big(\prod_t \tilde{V}_{8t+8}\Big)\Big(\prod_t \tilde{V}_{8t+4}\Big),
$$
$$
\tilde{U}^{o} = \Big(\prod_t \tilde{W}_{8t+11}\Big)\Big(\prod_t \tilde{W}_{8t+7}\Big)\Big(\prod_t \tilde{V}_{8t+9}\Big)\Big(\prod_t \tilde{V}_{8t+5}\Big),
\tag{54}
$$

where the subscript denotes the middle site of the subchain on which the local evolution operators acts, i.e. $\tilde{W}_t$ acts nontrivially on the sites $t-3$, $t-2$, $t-1$, $t$, $t+1$, $t+2$ and $t+3$. Graphically, this is represented by the following diagram,

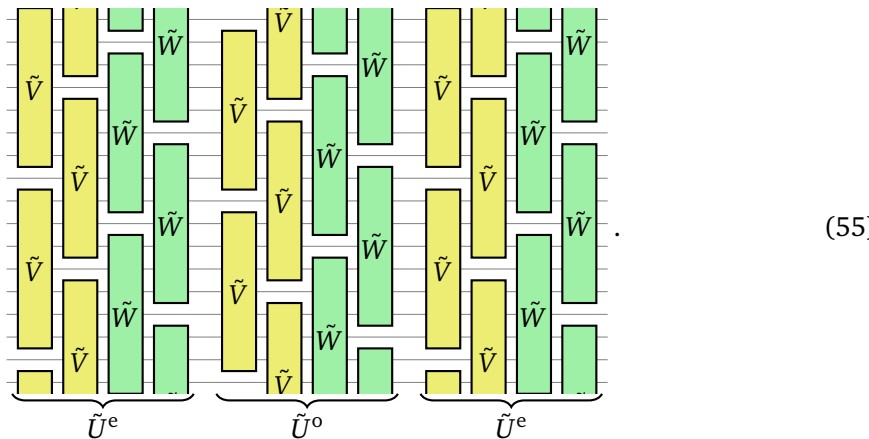

$$\tag{55}$$

The operators $\tilde{V}$, $\tilde{W}$ can be explicitly written in terms of the dual 3-site operators $\hat{U}$ by introducing the following 5-site projector $Q$,

$$
Q_{(s_1',s_2',s_3',s_4',s_5'),(s_1,s_2,s_3,s_4,s_5)} = \delta_{s_1,s_1'}\delta_{s_2,s_2'}\delta_{s_3,s_3'}\delta_{s_4,s_4'}\delta_{s_5,s_5'}
$$
$$
\cdot \left(1 - \delta_{s_2,0}\delta_{s_3,1}\delta_{s_1+s_4,1}\right)\left(1 - \delta_{s_4,0}\delta_{s_3,1}\delta_{s_2+s_5,1}\right). \tag{56}
$$

Then the 7-site gates can be expressed as a 3-site projected operator $\hat{U}$, sandwiched between two $P$ projectors on one and two $Q$ projectors on the other side,

$$
\tilde{V} \equiv \quad , \qquad \tilde{W} \equiv \quad , \tag{57}
$$

or equivalently

$$
\tilde{V}_t = Q_{t+1}Q_{t-1}\tilde{U}_t P_{t+1}P_{t-1}, \qquad \tilde{W}_t = P_{t+1}P_{t-1}\tilde{U}_t Q_{t+1}Q_{t-1}. \tag{58}
$$

These gates are deterministic on the restricted space of allowed time-configurations, since the following holds,

$$
\tilde{V}_t\tilde{V}_t^T = \tilde{W}_t^T\tilde{W}_t = Q_{t-1}P_t Q_{t+1}, \qquad \tilde{V}_t^T\tilde{V}_t = \tilde{W}_t\tilde{W}_t^T = P_{t-1}P_t P_{t+1}, \tag{59}
$$

where right-hand-sides are diagonal projection matrices with matrix elements that can only be 0 or 1. Therefore, to see that the space evolution is local and deterministic, we only have to show that the diagrams (55) and (52) are equivalent. The proof is provided in Appendix A.

# 5 Equilibrium time states

Similar ideas can be employed to find equilibrium time-states, i.e. the probability distributions of time-configurations under the assumption of the underlying system being in equilibrium (or stationary state). This provides an alternative derivation of the results of Ref. [32] that does not explicitly rely on the quasi-particle interpretation of dynamics. To simplify the discussion we first consider the infinite temperature state in 5.1, where we show how to use the properties of the local propagator (6) and its dual (41) to represent the time-state as a layer of observables squeezed between two vertically oriented MPSs with Schmidt rank 2. In 5.2 we generalize this to the class of equilibrium states introduced in section 2.2. In 5.3 we then reformulate the result in terms of a single MPS with Schmidt rank 4 and thus reproduce the main results of [32].

## 5.1 Maximum entropy state

In the case of maximum entropy (or infinite temperature) stationary state $\mathbf{p}_\infty = 2^{-2n}(\boldsymbol{\omega}^T)^{\otimes 2n}$, the vectorial form of the finite-size multi-time correlation function (30) exhibits a simple dia-

grammatic representation,

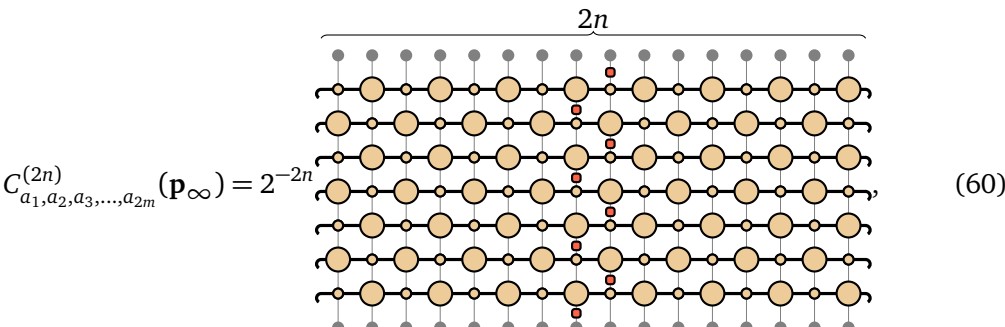

$$C^{(2n)}_{a_1,a_2,a_3,...,a_{2m}}(\mathbf{p}_\infty) = 2^{-2n} \qquad (60)$$

where the red squares represent (in general different) one-site observables and the grey circles denote one-site row (and column) vectors $\boldsymbol{\omega}$ (and $\boldsymbol{\omega}^T$). The local time-evolution operator $U$ is deterministic, which implies that it maps the three-site maximum entropy state into itself,

$$U(\boldsymbol{\omega}\otimes\boldsymbol{\omega}\otimes\boldsymbol{\omega})^T = (\boldsymbol{\omega}\otimes\boldsymbol{\omega}\otimes\boldsymbol{\omega})^T, \qquad (\boldsymbol{\omega}\otimes\boldsymbol{\omega}\otimes\boldsymbol{\omega})U = \boldsymbol{\omega}\otimes\boldsymbol{\omega}\otimes\boldsymbol{\omega}, \qquad (61)$$

This immediately allows us to simplify the diagrammatic expression by removing the gates from the top and bottom to obtain a light-cone structure,

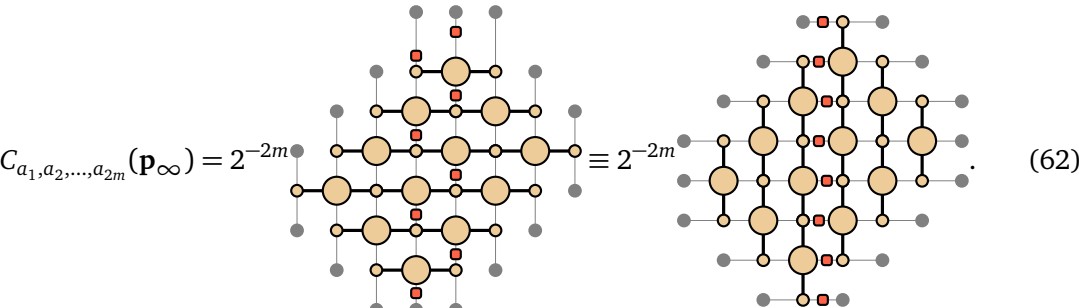

$$C_{a_1,a_2,...,a_{2m}}(\mathbf{p}_\infty) = 2^{-2m} \qquad \equiv 2^{-2m} \qquad . \qquad (62)$$

Note that the normalization factor is different with respect to (60), because of the normalization of vectors $\boldsymbol{\omega}$, namely $\boldsymbol{\omega}\boldsymbol{\omega}^T = 2$. The right hand diagram follows from the definition of (in general, non-deterministic) dual gates (41), and the fact that the observables can be understood as diagonal operators (see Appendix B for more details).

Up to now we made no assumption on the structure of dual evolution; the right-hand side of Eq. (62) follows from the deterministic nature of time evolution and the formal definition of the dual gate $\hat{U}$ (41). It holds for any deterministic 3-site propagator that nontrivially acts only on the middle site. [5] In our case the dual propagation is deterministic as well, therefore, in analogy to dual unitary circuits [11], we expect the diagram to further simplify. However, since the definition of local deterministic gates is rather involved (see Eq. (57)), the deterministic property cannot be directly used to reduce the diagram (62). Instead, we take advantage of the following two diagrammatic relations fulfilled by $\hat{U}$,

$$\qquad \equiv \qquad , \qquad \equiv \qquad . \qquad (63)$$

---

[5]To be more precise, the exact requirement is the validity of (61), which is satisfied by any bistochastic matrix.

We stress that even though these diagrams are conceptually similar to those used to prove the deterministic property of space evolution (cf. (94)), the precise relation between the two is not clear at present.

Because of $\hat{U}^T = \hat{U}$, also the left-right reversed diagrams hold. Additionally, since the observables are diagonal, they all commute with the three-site projector $P$. This, together with the relations (63), allows us to transform the correlation function $C_{a_1,a_2,\ldots,a_{2m}}(\mathbf{p}_\infty)$ into a diagram with only the two inner-most layers of dual gates left,

$$C_{a_1,a_2,a_3,\ldots,a_{2m}}(\mathbf{p}_\infty) = 2^{-2m} \qquad . \tag{64}$$

This can be put in a more convenient form by introducing left and right edge matrix product states, $(L|\mathbf{A}_1\mathbf{B}'_2\mathbf{A}_3\cdots\mathbf{A}_{2m-1}|R)$ and $(L|\mathbf{A}_2\mathbf{B}_3\mathbf{A}_4\cdots\mathbf{A}_{2m}|R)$, to replace the two remaining dual gate layers. The auxiliary space is 2-dimensional, with the following boundary vectors

$$\perp \equiv (L| = \begin{bmatrix} 1 & 1 \end{bmatrix}, \qquad \top \equiv |R) = \begin{bmatrix} 1 \\ 1 \end{bmatrix}. \tag{65}$$

The matrices $A_s$ are diagonal with one nonzero entry,

$$\ast \equiv \mathbf{A} \equiv \ast, \qquad A_0 = \begin{bmatrix} 1 & 0 \\ 0 & 0 \end{bmatrix}, \qquad A_1 = \begin{bmatrix} 0 & 0 \\ 0 & 1 \end{bmatrix}, \tag{66}$$

while the matrix elements of $B_s$, $B'_s$, diagrammatically represented by the squares

$$\square \equiv \mathbf{B}, \qquad \square \equiv \mathbf{B}', \qquad B_0 = B'_0 = \begin{bmatrix} 1 & 1 \\ 1 & 1 \end{bmatrix}, \qquad B_1 = B'_1 = \begin{bmatrix} 0 & 2 \\ 2 & 0 \end{bmatrix}, \tag{67}$$

are determined by requiring the following relations,

$$\equiv \quad , \qquad \equiv \quad , \qquad \equiv \quad , \qquad \equiv \quad . \tag{68}$$

Note that the first pair of diagrams follows from the second one due to $(L|A_0|R) = (L|A_1|R) = 1$. The correlation function can finally be rewritten as

$$C_{a_1,a_2,a_3,\ldots,a_{2m}}(\mathbf{p}_\infty) = 2^{-2m} \qquad . \tag{69}$$

The matrix-product-state (MPS) form of the time-state follows directly from here. However, before showing it explicitly, we first generalize the result to the class of equilibrium states $\mathbf{p}$, introduced in section 2.2.

## 5.2 Multi-time correlations for generic equilibrium states

To conveniently express multi-time correlations for the class of states **p** that can be expressed in a matrix product form (13), we introduce the diagrammatic notation for the MPS,

$$\mathbf{W} \equiv \blacktriangle, \qquad \mathbf{W}' \equiv \blacktriangledown, \qquad S \equiv \blacksquare, \tag{70}$$

which allows us to diagrammatically express finite-size correlation function as,

$$C^{(2n)}_{a_1,a_2,a_3,\ldots,a_{2m}}(\mathbf{p}) = \frac{1}{Z_{2n}}$$ 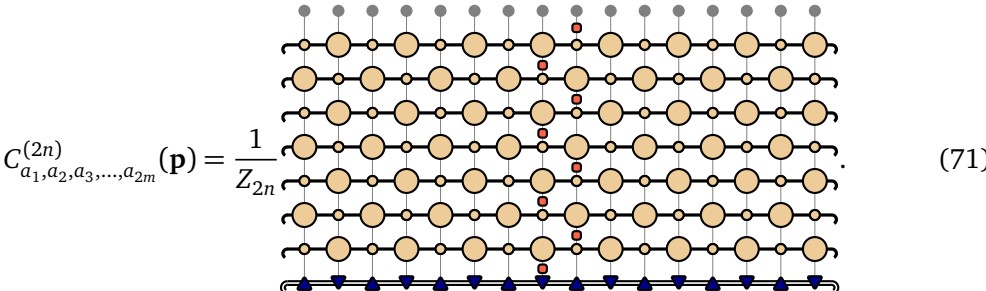 $$. \tag{71}$$

Similarly to the case of the maximum entropy state, the deterministic time evolution implies that the gates outside of the light-cone can be removed. To prove this, in addition to (61), we use the following three-site algebraic relations fulfilled by the state **p**,

$$\tag{72}$$

The first relation is a diagrammatic analogue of equation (17), while the others follow directly from it by using $U = U^{-1}$, and $S = S^{-1}$, as well as the simple mapping between **W** and **W**′. Additionally, we define $\langle l|$ and $|r\rangle$ as the left and right eigenvector of the matrix $(W'_0 + W'_1)(W_0 + W_1)$ that correspond to the leading eigenvalue $\lambda$, respectively

$$\tag{73}$$

Explicitly, $\lambda$ is the leading root of the following cubic equation

$$\lambda^3 - (1 + 3\xi\omega)\lambda^2 - (\xi + \omega + \xi\omega(1 - 3\xi\omega))\lambda - \xi\omega(1 - \xi\omega)^2 = 0, \tag{74}$$

and the leading eigenvectors $|r\rangle$, $\langle l|$ can be parametrized with $\lambda$, $\xi$ and $\omega$ as

$$
\begin{aligned}
|r\rangle &= \left[\xi(\lambda - \xi\omega + \omega) \quad (\lambda - \xi\omega)^2 - (\lambda + \omega) \quad \xi(\lambda - \xi\omega + \xi)\right], \\
\langle l| &= \left[(\lambda - \xi\omega)^2 - \xi\omega \quad \xi(\lambda - \xi\omega + \omega) \quad \omega(\lambda - \xi\omega + \xi)\right]^T.
\end{aligned}
\tag{75}
$$

The relations (72) immediately imply that the multi-time correlation function can be expressed in a form analogous to (62),

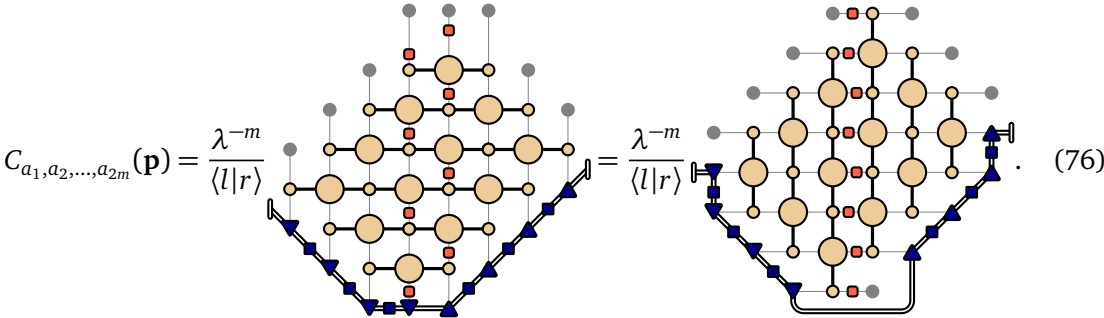

$$C_{a_1,a_2,\dots,a_{2m}}(\mathbf{p}) = \frac{\lambda^{-m}}{\langle l|r\rangle} \quad = \quad \frac{\lambda^{-m}}{\langle l|r\rangle} \quad . \tag{76}$$

One of the key ingredients in simplifying the circuit into a form analogous to (69) is a special factorization property of the equilibrium state $\mathbf{p}$, introduced in App. B of [32]. For any configuration of three consecutive edge sites $(s_1, s_2, s_3)$ the leftmost (rightmost) matrix can be absorbed into the left (right) boundary vector and replaced with a configuration-dependent prefactor. Namely, it is possible to define tensors of coefficients $\alpha_{s_1 s_2 s_3}$, $\alpha'_{s_1 s_2 s_3}$, $\beta_{s_1 s_2 s_3}$ and $\beta'_{s_1 s_2 s_3}$ so that the following holds,

$$W_{s_1} S W_{s_2} S W_{s_3} |r\rangle = \alpha_{s_1 s_2 s_3} W_{s_1} S W_{s_2} |r\rangle, \quad W'_{s_1} W_{s_2} S W_{s_3} |r\rangle = \beta_{s_1 s_2 s_3} W'_{s_1} W_{s_2} |r\rangle,$$
$$\langle l| W'_{s_1} S W'_{s_2} S W'_{s_3} = \alpha'_{s_1 s_2 s_3} \langle l| W'_{s_2} S W'_{s_3}, \quad \langle l| W'_{s_1} S W'_{s_2} W_{s_3} = \beta'_{s_1 s_2 s_3} \langle l| W'_{s_2} W_{s_3}. \tag{77}$$

As a consequence one is able to define vertically oriented left and right MPSs that replace layers of dual gates. These are analogous to the left and right edge states introduced for the maximum entropy case. The boundary vectors $(L|$ and $|R)$, as well as the matrices $A_s$ are defined in equations (65) and (66). The matrix elements of $B'_s$ are determined by the following relations,

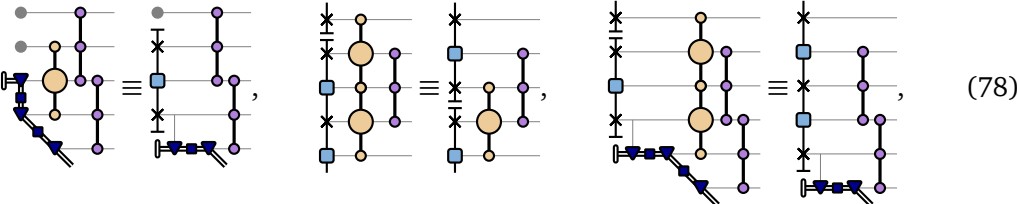

$$\tag{78}$$

while the matrices $B_s$ fulfill the analogous identities for the right edge,

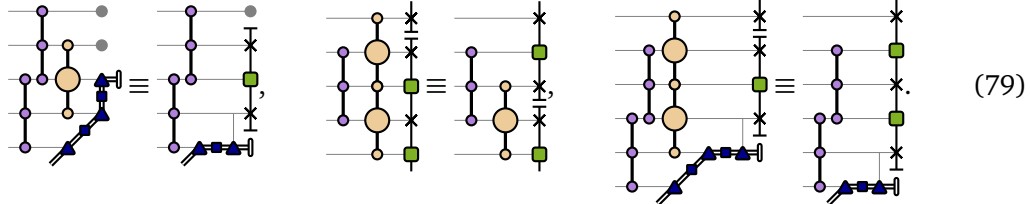

$$\tag{79}$$

The solution to these relations can be explicitly expressed in terms of tensors $\alpha_{s_1 s_2 s_3}$, $\alpha'_{s_1 s_2 s_3}$ as

$$B_0 = \begin{bmatrix} \alpha_{000} & \alpha_{001} \\ \alpha_{000} & \alpha_{001} \end{bmatrix}, \qquad B_1 = \begin{bmatrix} 0 & \alpha_{000} + \alpha_{001} \\ \alpha_{000} + \alpha_{001} & 0 \end{bmatrix},$$
$$B'_0 = \begin{bmatrix} \alpha'_{000} & \alpha'_{100} \\ \alpha'_{000} & \alpha'_{100} \end{bmatrix}, \qquad B'_1 = \begin{bmatrix} 0 & \alpha'_{000} + \alpha'_{100} \\ \alpha'_{000} + \alpha'_{100} & 0 \end{bmatrix}. \tag{80}$$

Additional details and the explicit values of the coefficient tensors are provided in Appendix C.

Using the relations (78) and (79), together with the observation that $(L|A_s|R) = 1$ for any $s$, namely

$$\overline{\underline{\mathbf{I}}} \equiv -\bullet, \qquad \overline{\underline{\mathbf{I}}} \equiv \bullet-, \tag{81}$$

the layers of dual gates in the diagram (76) can be removed one after another, until we are left with the innermost two layers squeezed between two vertical matrix product states,

$$C_{a_1,a_2,a_3,\ldots,a_{2m}}(\mathbf{p}) = \frac{\lambda^{-m}}{\langle l|r \rangle} \quad \text{[diagram]}. \tag{82}$$

To remove the last two layers, we note that the observables commute with all the projectors since they are diagonal in the same basis and structure of $\mathbf{B}$, $\mathbf{B}'$ implies that left (right) vertical states are invariant under projectors centered at odd (even) sites,

$$\text{[diagram]} \equiv \text{[diagram]}, \quad \text{[diagram]} \equiv \text{[diagram]}, \quad \text{[diagram]} \equiv \text{[diagram]}. \tag{83}$$

Additionally we have

$$\text{[diagram]} \equiv \text{[diagram]}, \quad \text{[diagram]} \equiv \text{[diagram]}. \tag{84}$$

These relations are analogous to the right-most diagrams from (78) and (79), and imply that the multi-time correlation function can be finally written as follows

$$C_{a_1,a_2,a_3,\ldots,a_{2m}}(\mathbf{p}) = \frac{\lambda^{-m}}{\langle l|r \rangle} \quad \text{[diagram]}. \tag{85}$$

In components Eq. (85) reads as

$$C_{a_1,\ldots,a_{2m}} = \sum_{s_1,s_2,s_3,\ldots,s_{2m}} \frac{\langle l|W'_{s_1} W_{s_2}|r \rangle}{\lambda^k \langle l|r \rangle} (L|A_{s_1} B'_{s_2} \cdots A_{s_{2m-1}}|R) \prod_{j=1}^{2m} a_j(s_j)(L|A_{s_2} B_{s_3} \cdots A_{s_{2m}}|R). \tag{86}$$

### 5.3 Matrix product representation of the time state

The above results can be expressed in terms of a time state, as defined in Section 2.4. The equilibrium time state $\mathbf{q} \in \mathbb{R}^{2^{2m}}$ corresponding to the equilibrium state $\mathbf{p}$ uniquely fixes multi-time correlation functions, which by definition implies

$$C_{a_1,a_2,a_3,\ldots,a_{2m}} = \sum_{s_1,s_2,s_3,\ldots,s_{2m}} q_{s_1 s_2 s_3 \ldots s_{2m}} \prod_{j=1}^{2m} a_j(s_j). \tag{87}$$

We can then read the probabilities of time-configurations $q_{s_1 s_2 \ldots s_{2m}}$ directly from (86) as

$$q_{s_1 s_2 s_3 \ldots s_{2m}} = \frac{\langle l | W'_{s_1} W_{s_2} | r \rangle}{\lambda^k \langle l | r \rangle} (L | A_{s_1} B'_{s_2} A_{s_3} \cdots A_{s_{2m-1}} | R)(L | A_{s_2} B_{s_3} A_{s_4} \cdots A_{s_{2m}} | R). \tag{88}$$

From here, an MPS representation is obtained by introducing matrices $\tilde{A}_s$, $\tilde{A}'_s$ that act on the 4-dimensional auxiliary space as

$$\tilde{A}_s = A_s \otimes B_s, \qquad \tilde{A}'_s = B'_s \otimes A_s, \tag{89}$$

and defining boundary vectors $|\tilde{R})$), $((\tilde{L}|$ as the solutions to the following relations,

$$\begin{aligned}
((\tilde{L} | \tilde{A}_{s_1} \tilde{A}'_{s_2} &= \frac{\langle l | W'_{s_1} W_{s_2} | r \rangle}{\langle l | r \rangle} \Big( (L | A_{s_1} B'_{s_2} \Big) \otimes \Big( (L | A_{s_2} \Big), \\
\tilde{A}_{s_1} \tilde{A}'_{s_2} | \tilde{R})) &= \Big( A_{s_1} | R) \Big) \otimes \Big( B_{s_1} A_{s_2} | R) \Big).
\end{aligned} \tag{90}$$

The time-state can thus be written in the matrix product form as

$$\mathbf{q} = \frac{1}{\lambda^k} ((\tilde{L} | \tilde{\mathbf{A}}_1 \tilde{\mathbf{A}}'_2 \tilde{\mathbf{A}}_3 \cdots \tilde{\mathbf{A}}'_{2m} | \tilde{R})). \tag{91}$$

As shown in Appendix D, this form of the time-state is equivalent to the MPS introduced in [32].

## 6 Conclusion

In this paper we have studied the properties of space evolution in Rule 54 reversible cellular automaton. We have shown that space translation of time configurations (i.e. configurations at the same position in the time direction) can be formulated as a reversible cellular automaton. In other words, the spatial dynamics can be expressed in terms of local deterministic maps with finite support. We have provided two different interpretations of local space evolution; as 7-site local deterministic maps, or equivalently, as a composition of non-deterministic 3-site gates and 3-site projectors onto the subspace of allowed configurations.

The result is interesting from two different points of view. On one hand, due to the existence of time states, the space dynamics of RCA54 can be studied as a novel solvable deterministic interacting model, where quasi-particles move with fixed velocities $\pm 1$ and undergo pairwise scattering. The main difference with respect to the usual (temporal) dynamics in RCA54 is the nature of two-body interaction, which speeds the particles up instead of slowing them down, i.e. it is repulsive rather than attractive. Arguably the more interesting perspective is to use the properties of the space dynamics to express nontrivial dynamical physical quantities. We have demonstrated this approach can be fruitful by finding an alternative derivation of the MPS form of *equilibrium time-states*, i.e. probability distributions that uniquely determine multi-time correlation functions at the same position.

The paper opens several interesting open questions. For instance, what is limit of this approach? It would be interesting to see whether the circuit picture can provide a new perspective on two-point spatio-temporal correlation functions [31] or time evolution of density matrices in the quantum version of the model [36]. This would provide a new perspective that does not explicitly rely on the quasi-particle interpretation of the dynamics, and is hence more robust and easier to generalize. Furthermore, one would like to understand whether RCA54 is an isolated example or it belongs to the bigger class of *dual reversible* cellular automata with a different but finite support of local evolution maps in space and time directions. Such a class would provide a generalization of dual unitary models [11] which could support richer transport properties, while many results would still be obtained exactly.

# Acknowledgements

KK thanks Bruno Bertini for insightful discussions and valuable comments on the manuscript.

**Funding Information** This work has been supported by the European Research Council under the Advanced Grant No. 694544 – OMNES, and by the Slovenian Research Agency (ARRS) under the Programme P1-0402.

# A  Equivalence between the deterministic and nondeterministic dual gates

The validity of (54) can be demonstrated graphically. First we recall that $P$ and $Q$ are projectors, i.e.

$$\tag{92}$$

Furthermore, all the local projectors commute and $\hat{U}$ commutes with all the projectors with which it shares at most one site. Explicitly, this implies the following diagrams,

$$\tag{93}$$

The last two properties needed for the proof are less trivial, but straightforward to check. Their diagrammatic form reads as

$$\tag{94}$$

Using these equalities, we can now easily show the equivalence between (55) and (52). First we express the propagator $\tilde{U}^{\mathrm{e}}$ from (55) in terms of gates $P$, $Q$ and $\hat{U}$,

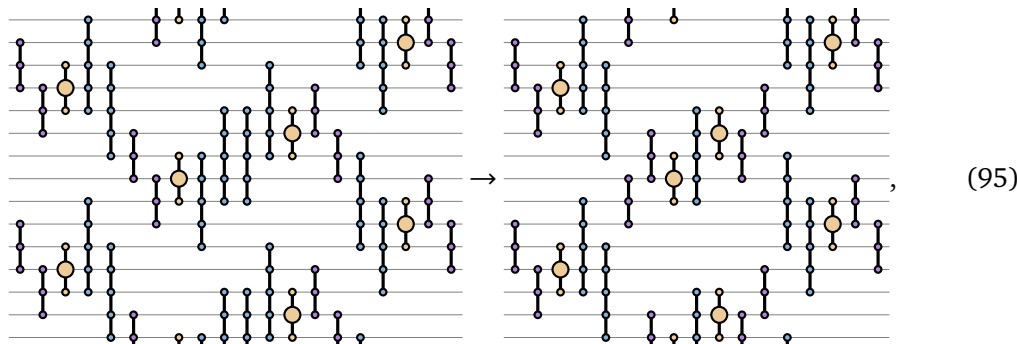

$$\tag{95}$$

where we simplified the diagram by using the fact $Q^2 = Q$ and commutation relations between the projectors and the gates (93). Using the second relation of (94) and moving around some of the commuting gates, we obtain the following,

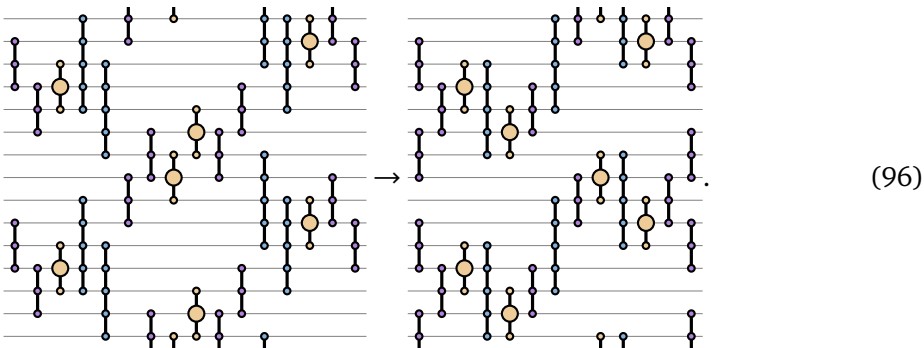

$$\tag{96}$$

Now we use the first equality of (94) and reposition the commuting gates so that we can again apply the second equality of (94),

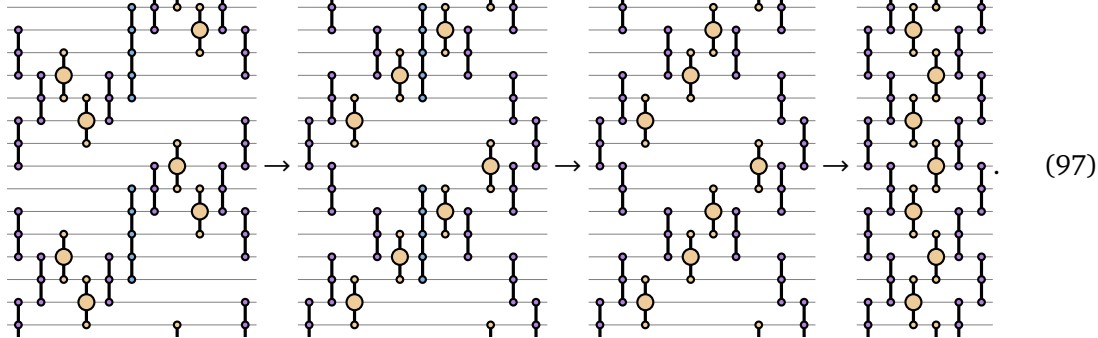

$$\tag{97}$$

In the final step we again rearranged the operators to obtain $\tilde{U}^{\mathrm{e}}$ as defined in (52). The same reasoning applies to $\tilde{U}^{\mathrm{o}}$, therefore (55) is equivalent to (52) and the dual propagation of RCA54 can be expressed in terms of local, deterministic 7-site gates on the reduced configuration space.

## B  Dual circuit representation of correlation functions

A convenient way to show the equivalence between the diagrams in Eq. (62) is to interpret the circuits as a 2-dimensional vertex model. Each line segment is either in the state $s = 0$

or $s = 1$, and the weights of vertices with large circles are given by the 3-site propagator $U$ as,

$$s_1 \quad \overset{s_4}{\underset{s_2}{\bigcirc}} \quad s_3 \quad \equiv U_{(s_1,s_4,s_3),(s_1,s_2,s_3)} = \hat{U}_{(s_2,s_1,s_4),(s_2,s_3,s_4)}. \tag{98}$$

The small circles force all the incoming lines to be in the same state,

$$s_1 \quad \overset{s_k}{\underset{s_2 \quad s_3}{\diagdown}} \quad \equiv \delta_{s_1,s_2} \delta_{s_2,s_3} \cdots \delta_{s_{k-1},s_k}, \tag{99}$$

where the weight is defined for any number $k \geq 2$ of intersecting lines. In particular, for $k = 2$ the diagram can be transformed into a straight line,

$$\overset{}{\underset{}{\multimap}} = \overset{}{\underset{}{\text{——}}}. \tag{100}$$

In this context, the one-site maximum entropy state $\omega$ corresponds to the sum of the line segment in states 0 and 1. This implies that we can always attach or remove lines connected to the maximum entropy state from the small circle, as long as at the end at least one such lines remains,

$$\succ\!\!\!\prec = \bullet\!\!-\!\!\circ\!\!\prec. \tag{101}$$

Using these relations, the equivalence of the diagrams from (62) can be recast as

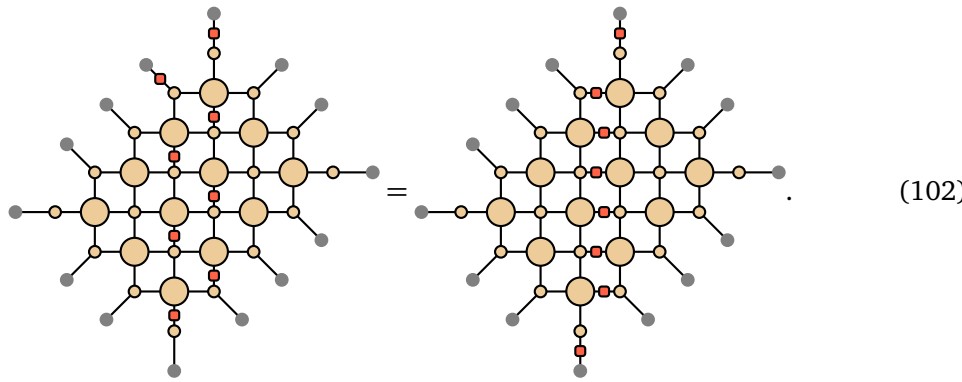

$$\tag{102}$$

This equality follows from the fact that the observables can be represented by diagonal one-site operators and can be therefore freely moved around the small circle,

$$\overset{\blacksquare}{\underset{}{\multimap}} = \overset{}{\underset{\blacksquare}{\multimap\!\!\blacksquare}} = \overset{}{\underset{}{\multimap}} = \blacksquare\!\!\multimap. \tag{103}$$

## C Factorization of the equilibrium state and the local few-site relations

We start by explicitly expressing coefficients $\alpha^{(\prime)}_{s_1 s_2 s_3}$, $\beta^{(\prime)}_{s_1 s_2 s_3}$ that satisfy factorization condition in equation (77). Solving the first two factorization relations we obtain the following solution,

$$
\begin{aligned}
&\alpha_{000} = \alpha_{100} = \alpha_{111} = 1, & &\alpha'_{000} = \alpha'_{001} = \alpha'_{111} = 1, \\
&\alpha_{001} = \alpha_{101} = \alpha_{110} = \frac{\xi(\lambda + \omega - \xi\omega)}{\lambda + \xi - \xi\omega}, & &\alpha'_{011} = \alpha'_{100} = \alpha'_{101} = \frac{\omega(\lambda + \xi - \xi\omega)}{\lambda + \omega - \xi\omega}, \\
&\alpha_{010} = \frac{\lambda + \xi - \xi\omega}{\lambda + \omega - \xi\omega}, & &\alpha'_{010} = \frac{\lambda + \omega - \xi\omega}{\lambda + \xi - \xi\omega}, \\
&\alpha_{011} = \frac{\omega(\lambda + \xi - \xi\omega)^2}{(\lambda + \omega - \xi\omega)^2}, & &\alpha'_{110} = \frac{\xi(\lambda + \omega - \xi\omega)^2}{(\lambda + \xi - \xi\omega)^2},
\end{aligned}
\tag{104}
$$

where we can immediately see that one set of parameters is transformed into another one by exchanging $\xi$ and $\omega$, and reversing the order of indices,

$$
\alpha'_{s_1 s_2 s_3} = \alpha_{s_3 s_2 s_1}\big|_{\xi \leftrightarrow \omega}.
\tag{105}
$$

Similarly, solving the bottom two equations we obtain

$$
\beta_{s_1 s_2 s_3} = \alpha_{s_1 s_2 s_3}, \qquad
\begin{aligned}
&\beta'_{000} = \beta'_{001} = \beta'_{110} = \alpha'_{000}, & &\beta'_{011} = \alpha'_{010}, \\
&\beta'_{010} = \beta'_{100} = \beta'_{101} = \alpha'_{100}, & &\beta'_{111} = \alpha'_{110}.
\end{aligned}
\tag{106}
$$

To demonstrate how the factorization property of the equilibrium state enables us to formulate the few-site relations (78) and (79), we first introduce the following notation for the basis vectors from $\mathbb{R}^{2^5}$,

$$
\mathbf{e}_{s_1 s_2 s_3 s_4 s_5} = \mathbf{e}_{s_1} \otimes \mathbf{e}_{s_2} \otimes \mathbf{e}_{s_3} \otimes \mathbf{e}_{s_4} \otimes \mathbf{e}_{s_5}, \qquad
\mathbf{e}_0 = \begin{bmatrix} 1 \\ 0 \end{bmatrix}, \qquad
\mathbf{e}_1 = \begin{bmatrix} 0 \\ 1 \end{bmatrix}.
\tag{107}
$$

Now we can express the first identity from Eq. (78) in explicit component form as,

$$
\begin{aligned}
&\sum_{s_1, s_2, s_3, s_4, s_5} \mathbf{e}_{s_1 s_2 s_3 s_4 s_5} \hat{U}_3 P_2 P_4 \cdot \langle l | W'_{s_3} S W'_{s_2} S W'_{s_1} \\
&= \sum_{s_1, s_2, s_3, s_4, s_5} \mathbf{e}_{s_1 s_2 s_3 s_4 s_5} \hat{U}_3 P_2 P_4 \cdot \alpha'_{s_3 s_2 s_1} \langle l | W'_{s_2} S W'_{s_1} \\
&= \sum_{s_1, s_2, s_3, s_4, s_5} \mathbf{e}_{s_1 s_2 s_3 s_4 s_5} P_2 P_4 \cdot (l | A_{s_2} B'_{s_3} A_{s_4} | r) \langle l | W'_{s_2} S W'_{s_1},
\end{aligned}
\tag{108}
$$

where to get from the first to the second line, we used the first of the factorization conditions (77). Note that $\mathbf{B}$ and $\mathbf{B}'$ satisfy an even stronger condition, where we can remove the sum over $s_1$ and $s_2$. Namely,

$$
\sum_{s_3, s_4, s_5} \mathbf{e}_{s_1 s_2 s_3 s_4 s_5} \hat{U}_3 P_2 P_4 \cdot \alpha'_{s_3 s_2 s_1} = \sum_{s_3, s_4, s_5} \mathbf{e}_{s_1 s_2 s_3 s_4 s_5} P_2 P_4 \cdot (l | A_{s_2} B'_{s_3} A_{s_4} | r).
\tag{109}
$$

# D  Matrix-product form of multi-time correlation functions

To see that the MPS representation (91) is equivalent to time-states introduced in [32] we first explicitly spell out the matrices $\tilde{A}_s$, $\tilde{A}'_s$ which by definition (89) take the following form,

$$
\tilde{A}_0 = \begin{bmatrix} \alpha_{000} & \alpha_{001} & 0 & 0 \\ \alpha_{000} & \alpha_{001} & 0 & 0 \\ 0 & 0 & 0 & 0 \\ 0 & 0 & 0 & 0 \end{bmatrix}, \qquad \tilde{A}_1 = \begin{bmatrix} 0 & 0 & 0 & 0 \\ 0 & 0 & 0 & 0 \\ 0 & 0 & 0 & \alpha_{000} + \alpha_{001} \\ 0 & 0 & \alpha_{000} + \alpha_{001} & 0 \end{bmatrix},
$$
$$
\tilde{A}'_0 = \begin{bmatrix} \alpha'_{000} & 0 & \alpha'_{100} & 0 \\ 0 & 0 & 0 & 0 \\ \alpha'_{000} & 0 & \alpha'_{100} & 0 \\ 0 & 0 & 0 & 0 \end{bmatrix}, \qquad \tilde{A}'_1 = \begin{bmatrix} 0 & 0 & 0 & 0 \\ 0 & 0 & 0 & \alpha'_{000} + \alpha'_{100} \\ 0 & 0 & 0 & 0 \\ 0 & 0 & \alpha'_{000} + \alpha'_{100} & 0 \end{bmatrix},
$$

(110)

while the boundary vectors $((\tilde{L}|$ and $|\tilde{R}))$ that solve equation (90) can be after some straightforward algebraic manipulation expressed as

$$
((L| = \frac{\alpha'_{000} + \alpha'_{100}}{1 + \frac{\alpha_{001}}{\alpha_{000} + \alpha_{001}} + \frac{\alpha'_{100}}{\alpha'_{000}\alpha'_{100}}} \left[ \frac{\alpha'_{000}}{\alpha'_{000} + \alpha'_{100}} \quad \frac{\alpha'_{100}}{\alpha'_{000} + \alpha'_{100}} \quad \frac{\alpha'_{100}}{\alpha'_{000} + \alpha'_{100}} \quad \frac{\alpha_{001}}{\alpha_{000} + \alpha_{001}} \right],
$$
$$
|R)) = \frac{1}{\alpha'_{000} + \alpha'_{100}} \begin{bmatrix} 1 & 1 & 1 & 1 \end{bmatrix}^T.
$$

(111)

Additionally, we note that the product $(\alpha_{000} + \alpha_{001})(\alpha'_{000} + \alpha'_{100})$ is equal to the leading eigenvalue $\lambda$ of $(W'_0 + W'_1)(W_0 + W_1)$,

$$
\lambda = (\alpha_{000} + \alpha_{001})(\alpha'_{000} + \alpha'_{100}). \tag{112}
$$

Equipped by these relations, it is easy to see that it is possible to introduce linear maps $Q$, $U$, $V$,

$$
U = \begin{bmatrix} 1 & 0 & 0 & -\frac{\alpha_{001}}{\alpha_{000}} \\ 1 & 0 & 0 & 1 \\ 0 & 0 & 1 & 0 \\ 0 & 1 & 0 & 0 \end{bmatrix}, \qquad V = \begin{bmatrix} 1 & 0 & 0 & -\frac{\alpha'_{100}}{\alpha_{000}} \\ 0 & 0 & 1 & 0 \\ 1 & 0 & 0 & 1 \\ 0 & 1 & 0 & 0 \end{bmatrix}, \qquad Q = \begin{bmatrix} 1 & 0 & 0 & 0 \\ 0 & 1 & 0 & 0 \\ 0 & 0 & 1 & 0 \end{bmatrix}, \quad (113)
$$

so that the following holds for any $s_1, s_2 \in \{0, 1\}$,

$$
\begin{aligned}
\tilde{A}_{s_1} U Q^T Q U^{-1} \tilde{A}'_{s_2} &= \tilde{A}_{s_1} \tilde{A}'_{s_2}, & \tilde{A}'_{s_1} V Q^T Q V^{-1} \tilde{A}_{s_2} &= \tilde{A}'_{s_1} \tilde{A}_{s_2}, \\
((\tilde{L}| V Q^T Q V^{-1} \tilde{A}_{s_1} &= ((\tilde{L}| \tilde{A}_{s_1}, & \tilde{A}'_{s_1} V Q^T Q V^{-1} |\tilde{R})) &= \tilde{A}'_{s_1} |\tilde{R})).
\end{aligned}
$$

(114)

This implies that the state (91) can be equivalently represented by an MPS with a 3-dimensional auxiliary space

$$
\mathbf{q} = \langle x_L | \mathbf{X}_1 \mathbf{X}'_2 \mathbf{X}_3 \cdots \mathbf{X}'_{2m} | x_R \rangle, \tag{115}
$$

where the new matrices $X_s$, $X'_s$ and boundary vectors $\langle x_L|$, $|x_R\rangle$ are defined as

$$
\begin{aligned}
X_s &= \frac{1}{\alpha_{000} + \alpha_{001}} Q V^{-1} \tilde{A}_s U Q^T, & X'_s &= \frac{1}{\alpha'_{000} + \alpha'_{001}} Q V^{-1} \tilde{A}'_s U Q^T, \\
\langle x_L| &= \frac{1}{\alpha'_{000} + \alpha'_{100}} ((\tilde{L}| V Q^T, & |x_R\rangle &= (\alpha'_{000} + \alpha'_{100}) Q V^{-1} |\tilde{R})),
\end{aligned}
$$

(116)

which implies the following explicit form,

$$
X_0 = \begin{bmatrix} \frac{\alpha'_{000}}{\alpha'_{000}+\alpha'_{100}} & 0 & 0 \\ 0 & 0 & 0 \\ 1 & 0 & 0 \end{bmatrix}, \qquad X_1 = \begin{bmatrix} 0 & \frac{\alpha'_{100}}{\alpha'_{000}+\alpha'_{100}} & 0 \\ 0 & 0 & 1 \\ 0 & 0 & 0 \end{bmatrix},
$$

$$
X'_0 = \begin{bmatrix} \frac{\alpha_{000}}{\alpha_{000}+\alpha_{001}} & 0 & 0 \\ 0 & 0 & 0 \\ 1 & 0 & 0 \end{bmatrix}, \qquad X'_1 = \begin{bmatrix} 0 & \frac{\alpha_{100}}{\alpha_{000}+\alpha_{001}} & 0 \\ 0 & 0 & 1 \\ 0 & 0 & 0 \end{bmatrix}, \tag{117}
$$

$$
\langle x_L | = \frac{1}{1 + \frac{\alpha_{001}}{\alpha_{000}+\alpha_{001}} + \frac{\alpha'_{100}}{\alpha'_{000}+\alpha'_{100}}} \begin{bmatrix} 1 & \frac{\alpha_{001}}{\alpha_{000}+\alpha_{001}} & \frac{\alpha'_{100}}{\alpha'_{000}+\alpha'_{100}} \end{bmatrix},
$$

$$
| x_R \rangle = \begin{bmatrix} 1 & 1 & 1 \end{bmatrix}^T.
$$

Finally, to see that this parametrization coincided with the MPS from Ref. [32], one needs to only express parameters $\frac{\alpha_{001}}{\alpha_{000}+\alpha_{001}}$ and $\frac{\alpha'_{100}}{\alpha'_{000}+\alpha'_{100}}$ in terms of $\xi$, $\omega$ and $\lambda$.

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
