# Peer review of "Space-like dynamics in a reversible cellular automaton"

_SciPost Physics, doi:SciPost Phys. Core 2, 010 (2020)_

## Round 1 · Referee Report · Anonymous · 2020-4-30

Report

Space-like dynamics in a reversible cellular automaton

This is an impressively complete treatment of a classical reversible cellular automaton with the interesting property of being “dual reversible”. The authors define this to mean that exchanging the space and time directions gives another well-defined reversible cellular automaton. The authors work out the definition of this dual automaton, giving a couple of equivalent formulations, and use their constructions to rederive a general formula for time correlations.

The paper seems to me highly suitable for SciPost essentially in its current state. The results are interesting and the presentation is clear and precise. The initial discussion gives a clear motivating physical picture in terms of solitons. There are interesting aspects to how the duality works out, for example the need to project to a subspace of states and the existence of the two alternative formulations (as well as, more simply, the sign change in the soliton interactions). Using the circuit formalism to reduce arbitrary multi-time correlations at a site to a simple formula is also notable. For a technical paper with many formulas it is relatively readable because excellent use is made of diagrammatic notation.

My comments are minor and about presentation rather than content.

Abstract. The authors use the term “dual reversible” without explanation. This may not be sufficiently clear. For example the reader may think that a self-duality of the model is implied, which it is not: the model is dual to a different model.

Bottom of page 2. This paragraph is discussing classical models with continuous degrees of freedom. This should be made more explicit, since in the next paragraph the authors wish to contrast with models of discrete degrees of freedom.

Page 3, first complete paragraph. If the reader googles “Rule 54 cellular automaton” they will find references to the model defined by Wolfram in 1983. Here the same terminology is used for a model which is almost the same but not quite. If I understand correctly, the local update rule is the same, but the pattern in which updates are applied is distinct (and this difference is crucial for the physical properties). It would be useful to have a sentence clarifying the distinction, so that the reader is aware that there are two models referred to by the same name.

Eq 11. The notation for the subscripts on W seems not to be quite consistent with the discussion that follows.

Could the discussion around (14) be made more digestible? Why is S called a delimiter?

Eqs 39, 40. The reader may need some clarification of the notation, which does not quite follow standard tensor network conventions. Perhaps a reference to Appendix B could be useful.

Sec 5 this is a sequence of explicit calculations. Could some brief higher-level discussion be helpful to readers who do not follow the calculations line by line?

Can the authors comment further on the space of dual-reversible models? Are there nontrivial examples that are self-dual rather than being dual to other reversible models?

---

## Round 1 · Referee Report · Anonymous · 2020-5-4

Strengths

1-The manuscript constructs the space-like dynamics of a cellular automaton model. That is, a deterministic rule is derived, by which trajectories of the automaton can be generated by iterating along the space-like direction of the model, instead of iterating over successive time steps (as would conventionally be the case). This is an interesting result in itself.

2- The manuscript explains how this space-like dynamics can be understood via a circuit representation. This representation is applied, to derive a matrix-product representation of the probabilities of different "time configurations", which are defined by considering the time-evolution of the model at a single spatial point (more precisely, at two adjacent points). This representation was obtained already in a previous work, but the alternative derivation is useful, particularly because there are possibilities to apply this new method more generally (in other models).

3-The arguments in the manuscript are laid out clearly and in detail. The methods are technical in places but this does not hide the key physical insights.

Weaknesses

1-Despite point 3 mentioned in "strengths" there are some places where the arguments could be clarified.

Report

I do not have any significant concerns about the validity of these results, especially since they recover previous results for the same model. Most of the manuscript succeeds in communicating a fairly technical analysis in a very clear way. However, I think a few improvements in this area can help to make the manuscript more accessible. This would be useful if other scientists want to apply the same methodology in other systems.

I also have a suggestion as to how the "allowed space of time-configurations" can be interpreted, see requested changes, below.

Requested changes

These requested changes are quite detailed but they should be minor.

1-Above eq(26), where the rules are stated for allowed "time configurations". It seems to me that these rules can be enforced by considering the sequence '110' as a "rod-like particle" of size 3. Then the allowed configurations are exactly those where these rod-like particles do not overlap. (The spaces between rod-like particles are filled with zeros.) I think this point should be mentioned. I suspect that the space-like dynamics can be expressed quite simply in terms of the behaviour of these rod-like particles, the authors may want to check if this is the case. However, this may be beyond the scope of the current work. (It would be interesting to understand if also the equilibrium (matrix product) states have a nice representation in terms of the rods.)

2- Equ(33,34), it is not clear what logical process is represented by the red arrows. As a result, it took me some time to work out what is the line of argument. [In (33), I think the arrows are a logical process of inference on a fixed set of s-variables, but in (34) they indicate evolution of space-like dynamics.] I also think that it would be sensible to use explicitly the rule for allowed configurations (point 1 above) to reduce the number of gray squares in these diagrams (in several cases, the state of a gray square can be fixed by the constraint on allowed configurations). This last point would be especially useful in eq(35) where (by my reckoning) it would reduce he number of gray (uncertain) squares very considerably. [I also think that this will help to reveal the physical features of the dynamics.] Overall, a bit more clarification would be helpful throughout this section, which is central to the paper.

3- As a non-expert in circuit representations, I suggest some clarification is needed as to what exactly is represented by (38,40) and in what sense (38) "can be replaced by" (40). Is the assumption of doubly periodic boundaries required for (38) to be replaced by (40)? (why?) Without full definitions, I am not sure why the top left circle is large in (40) but small in (38), or if this is irrelevant because of periodic boundaries. I also suggest that a diagram similar to the leftmost picture in eq(4) may be useful to motivate (39), in which the positioning of the various indices may seem strange on a first glance.

4- A few small points. Below eq(13), perhaps clarify in what sense xi,omega are "spectral" parameters? Below (28) the fact that 2.floor(n/2) corresponds to a point far from the spatial boundaries might be noted (the factor of 2 in front may be unexpected but (I think) is present because the "system size" is 2n and not n). Below (60), it is noted that this equation depends only on "the formal definition" of \hat{U}, it is not clear to me where is this definition, is it eq.(96)?

---

## Round 2 · Author Response

We thank both the referees for the careful reading of the manuscript and useful suggestions. Below is a response to referee remarks.

Reply to Referee 1:

1-"Abstract. The authors use the term “dual reversible” without explanation. This may not be sufficiently clear. For example the reader may think that a self-duality of the model is implied, which it is not: the model is dual to a different model."

Response: We modified the wording of the abstract slightly to clear the possible confusion.

2-"Bottom of page 2. This paragraph is discussing classical models with continuous degrees of freedom. This should be made more explicit, since in the next paragraph the authors wish to contrast with models of discrete degrees of freedom."

Response: We added a remark.

3-"Page 3, first complete paragraph. If the reader googles “Rule 54 cellular automaton” they will find references to the model defined by Wolfram in 1983. Here the same terminology is used for a model which is almost the same but not quite. If I understand correctly, the local update rule is the same, but the pattern in which updates are applied is distinct (and this difference is crucial for the physical properties). It would be useful to have a sentence clarifying the distinction, so that the reader is aware that there are two models referred to by the same name."

Response: We extended the footnote on pg. 3 to explain this point.

4&5-"Eq 11. The notation for the subscripts on W seems not to be quite consistent with the discussion that follows. Could the discussion around (14) be made more digestible? Why is S called a delimiter?"

Response: We removed the unnecessary jargon ("delimiter matrix") and restructured the discussion of stationary states.

6-"Eqs 39, 40. The reader may need some clarification of the notation, which does not quite follow standard tensor network conventions. Perhaps a reference to Appendix B could be useful."

Response: We added a sentence for clarification and a reference to Appendix B. We also corrected a mistake in eq. (40) (now eq. 42) - as noted by Referee 2.

7-"Sec 5 this is a sequence of explicit calculations. Could some brief higher-level discussion be helpful to readers who do not follow the calculations line by line?"

Response: We reformulated the beginning of the Section 5 and added a short summary of the section to make it easier for the reader to follow.

8-"Can the authors comment further on the space of dual-reversible models? Are there nontrivial examples that are self-dual rather than being dual to other reversible models?"

Response: Although undoubtedly an interesting question, we feel that we cannot adequately address it in sufficient generality at the moment. An example of a self-dual RCA is RCA150, which corresponds to changing the update rule (eq. 2) by removing the term s1.s3. This can be understood as a "noninteracting" version of the model and we expect physics to be in some sense simpler (in particular, time-states in Section 5 trivialize since all the dual gates can be removed).

Reply to Referee 2:

1-"Above eq(26), where the rules are stated for allowed "time configurations". It seems to me that these rules can be enforced by considering the sequence '110' as a "rod-like particle" of size 3. Then the allowed configurations are exactly those where these rod-like particles do not overlap. (The spaces between rod-like particles are filled with zeros.) I think this point should be mentioned. I suspect that the space-like dynamics can be expressed quite simply in terms of the behaviour of these rod-like particles, the authors may want to check if this is the case. However, this may be beyond the scope of the current work. (It would be interesting to understand if also the equilibrium (matrix product) states have a nice representation in terms of the rods.)"

Response: We thank the referee for an interesting observation, which is an indeed an 67 excellent proposal for future work.

2-"Equ(33,34), it is not clear what logical process is represented by the red arrows. As a result, it took me some time to work out what is the line of argument. [In (33), I think the arrows are a logical process of inference on a fixed set of s-variables, but in (34) they indicate evolution of space-like dynamics.] I also think that it would be sensible to use explicitly the rule for allowed configurations (point 1 above) to reduce the number of gray squares in these diagrams (in several cases, the state of a gray square can be fixed by the constraint on allowed configurations). This last point would be especially useful in eq(35) where (by my reckoning) it would reduce he number of gray (uncertain) squares very considerably. [I also think that this will help to reveal the physical features of the dynamics.] Overall, a bit more clarification would be helpful throughout this section, which is central to the paper."

Response: We rewrote a part of the discussion and made the diagrams more explicit.

3-"As a non-expert in circuit representations, I suggest some clarification is needed as to what exactly is represented by (38,40) and in what sense (38) "can be replaced by" (40). Is the assumption of doubly periodic boundaries required for (38) to be replaced by (40)? (why?) Without full definitions, I am not sure why the top left circle is large in (40) but small in (38), or if this is irrelevant because of periodic boundaries. I also suggest that a diagram similar to the leftmost picture in eq(4) may be useful to motivate (39), in which the positioning of the various indices may seem strange on a first glance."

Response: We thank the referee for bringing up the "typo" in eq. (40) (now eq. (42))- indeed both circles are supposed to have the same size. We added a short sentence of clarification and a reference to Appendix B, where an alternative viewpoint is mentioned.

4a-"A few small points. Below eq(13), perhaps clarify in what sense xi,omega are "spectral" parameters?"

Response: We reformulated the discussion of stationary states and provided another physical interpretation of the equilibrium states. We no longer refer to these parameters as "spectral".

4b-"Below (28) the fact that 2.floor(n/2) corresponds to a point far from the spatial boundaries might be noted (the factor of 2 in front may be unexpected but (I think) is present because the "system size" is 2n and not n)."

Response: We added a remark.

4c-"Below (60), it is noted that this equation depends only on "the formal definition" of \hat{U}, it is not clear to me where is this definition, is it eq.(96)?"

Response: We added a reference to eq. (39) at that place (now eq. (41)).

---

## Round 2 · List of Changes

1- The wording of the abstract is slightly modified (3rd sentence)

2- Added a short remark at the end of 3rd paragraph in Introduction

3- Extended a footnote on pg. 3

4- Reformulated the discussion of stationary states in 2.2

5- Added a remark after eq. (29)

6- Reformulated Section 3, after eq. (33)

7- Added a remark and reference to AppB between eqs (40) and (42) and corrected eq. 42

8- Added a short summary of results to the beginning of Section 5

9- Added a reference to eq (41) in the paragraph before eq (63)

---

## Editorial Decision

published